# Cross-site transportability of an explainable artificial intelligence model for acute kidney injury prediction

Xing Song [1], Alan S. L. Yu [2], John A. Kellum [3], Lemuel R. Waitman[1], Michael E. Matheny [4,5], Steven Q. Simpson[6], Yong Hu [7]✉ & Mei Liu [1]✉

Artificial intelligence (AI) has demonstrated promise in predicting acute kidney injury (AKI), however, clinical adoption of these models requires interpretability and transportability. Non-interoperable data across hospitals is a major barrier to model transportability. Here, we leverage the US PCORnet platform to develop an AKI prediction model and assess its transportability across six independent health systems. Our work demonstrates that cross-site performance deterioration is likely and reveals heterogeneity of risk factors across populations to be the cause. Therefore, no matter how accurate an AI model is trained at the source hospital, whether it can be adopted at target hospitals is an unanswered question. To fill the research gap, we derive a method to predict the transportability of AI models which can accelerate the adaptation process of external AI models in hospitals.

[1] Division of Medical Informatics, Department of Internal Medicine, University of Kansas Medical Center, Kansas City, KS, USA. [2] Division of Nephrology and Hypertension and the Kidney Institute, School of Medicine, University of Kansas Medical Center, Kansas City, KS, USA. [3] Center for Critical Care Nephrology, Department of Critical Care Medicine, University of Pittsburgh School of Medicine, Pittsburgh, PA, USA. [4] Department of Biomedical Informatics, Department of Medicine, Department of Biostatistics, Vanderbilt University School of Medicine, Nashville, TN, USA. [5] Geriatrics Research Education and Clinical Care Center, Tennessee Valley Healthcare System VA, Nashville, TN, USA. [6] Division of Pulmonary, Critical Care, and Sleep Medicine, Department of Internal Medicine, University of Kansas Medical Center, Kansas City, KS, USA. [7] Big Data Decision Institute, Jinan University, Guangzhou, Guangdong, China. ✉email: yonghu@jnu.edu.cn; meiliu@kumc.edu

Acute kidney injury (AKI) is a potentially life-threatening clinical syndrome for which there is no effective treatment other than supportive care and dialysis. AKI complicates the course of treatment and worsens outcomes of significant numbers of hospitalized-patients, i.e., 10–15% of all inpatients[1] and more than 50% of the critical care patients[2]. In-hospital mortality among patients with AKI is high, with as much as a sevenfold increased mortality risk compared with patients without AKI[1]. Patients who survive an episode of AKI are also at increased risk for long-term adverse health outcomes such as the development of cardiovascular disease or progression to chronic kidney disease or end-stage renal disease[3]. Biomarkers to predict risk, diagnosis, prognosis, and therapeutic responses in patients with AKI are being identified and evaluated in clinical studies. However, the only FDA-approved biomarker test, Nephrocheck (Astute Medical, San Diego CA), is indicated for critically ill patients, whereas substantial proportion of patients developing AKI are cared for outside the intensive care units.

Recent availability of electronic health record (EHR) data and advances in artificial intelligence (AI) have sparked growing interest in machine learning-based risk prediction models for healthcare[4,5]. The emerging applications of machine-learning algorithms for prediction of patient outcomes using EHR data have demonstrated promise in helping physicians to anticipate future events at an expert level, and in some cases surpassing the performance of clinicians[6]. Traditional machine learning-based AKI prediction models have achieved an area under the receiver operating characteristic curve (AUROC) ranging 0.71–0.80 in derivation studies, 0.66–0.80 in internal validation studies, and 0.65–0.71 in external validation studies[7–9]. A recent study reported in Nature[10] by Google DeepMind applied deep learning on EHR data for AKI prediction and showed that their model could predict 55.8% of all inpatient episodes of AKI with lead times up to 48 h. However, the deep learning model was trained on medical records of mostly Caucasian male patients admitted to the US Department of Veterans Affairs healthcare system, lacking independent validation on a general population from other health systems[11]. Koyner et al.[8] developed and recently externally validated[9] their AKI prediction model in patients treated at hospitals in Illinois using gradient boosting machine, which achieved an AUROC above 0.85 for AKI stage 2 prediction within 48 h.

Before prediction models can be implemented in clinical practice, their transportability or ability to produce accurate predictions on new sets of patients from different settings, e.g., clinical settings, geographical locations, or time periods, must be validated[12]. Failure of a prediction model to transport well across settings indicates that the model cannot be readily implemented in clinical practice for new patients[13]. Model transportability is often evaluated by independent validation using external data resources, e.g., AKI patients from different health systems. Major barriers to independent validation include patient heterogeneity, clinical process variability, and EHR configuration and data warehouse heterogeneity leading to non-interoperable databases across hospitals. Prediction model performance can vary as a result of changing outcome rates, shifting patient populations, and evolving clinical practice[14]. Most available AKI prediction models require considerable manual curation effort in pre-selecting predictor variables; however, certain variables may be only available to a single hospital or clinical unit that hinders the transportability of a model across different settings.

To ensure data harmonization and consistency across health systems, many healthcare institutions have transformed data contained in their EHR into a common representation (terminologies, vocabularies, coding schemes) as well as a common format (common data model or CDM)[15,16]. In 2010, the United States Congress authorized the creation of the non-profit Patient-Centered Outcomes Research Institute (PCORI) who launched PCORnet[17] in 2014 to support the development of an ecosystem for conducting patient-centered outcomes research faster using EHR across hundreds of health systems in the US. In this study, we utilized EHR data in the Greater Plains Collaborative (GPC)[18], a PCORnet Clinical Data Research Network consisting of twelve independent health systems in nine US states, to build a scalable and interpretable machine-learning model to continuously calculate AKI risk within the next 48 h for all inpatients from their admission to the hospital until discharge. The underlying learning algorithm is gradient boosting with decision trees implemented in a discrete-time survival framework (*DS-GBT*) that operates sequentially over individual EHRs with independent right-censoring (see Supplementary Fig. 1). GPC's geographical reach across nine central US states is advantageous for testing generalizability of research findings. We assessed the transportability of the AKI prediction model on patients from six GPC sites and examined risk factor patterns within and across populations. Finally, we derived a statistical method to predict the transportability of prediction models when transported from one hospital (source) to another (target) without full disclosure of the target hospital data, which can accelerate the adaptation process of external AI models in hospitals.

## Results

**Development and validation of AKI prediction model in source health system.** From the source health system, we collected a sample of 153,821 eligible inpatient encounters between 1 January 2010 and 31 December 2018 with various lengths of stay. Each individual patient had a mean of 67 (SD = 46) clinical facts per day with a total number of 1,064,619 encounter-days. After automated data curation, the final dataset contained 38,920 unique variables with 142,167,783 distinct observations. Among them, only 1933 unique variables (i.e., the common set) shared across the six participating sites. Cohort identification and automated data curation procedures are described in "Methods". We held-out inpatient encounters occurring after 1 January 2017 as the temporal validation set (27,603 encounters) and randomized the remaining encounters between 1 January 2010 and 31 December 2016, across derivation (70%), calibration (15%), and internal validation (15%) sets. The AKI prediction model was trained and optimized using the derivation set and evaluated on both internal and temporal validation sets.

Figure 1 illustrates the model performance, measured by AUROC, in predicting inpatient risks of developing different stages of AKI within the next 48 h using all collected predictor variables or with all forms of serum creatinine (SCr) and blood urea nitrogen (BUN) removed. Model performance across various subgroups, i.e., stratified by baseline SCr, patient age at admission, days since admission, and type of validation, is also shown in Fig. 1. Temporal hold-out validation mimics the prospective evaluation of the model on future unseen hospital encounters, which demonstrated an overall AUROC of 0.76 [95% CI, 0.75–0.78] for any AKI prediction within 48 h, 0.81 [95% CI, 0.76–0.86] for predicting at least AKI stage 2 (moderate-to-severe AKI), and 0.87 [95% CI, 0.78–0.93] for predicting AKI stage 3 (severe AKI). The overall area under the precision-recall curve (AUPRC) was 0.14 [95% CI, 0.08–0.23] for predicting moderate-to-severe AKI in 48 h, and achieved 0.23 [95% CI, 0.08–0.42] for severe AKI prediction (Supplementary Fig. 2). The model excluding SCr and BUN in all forms showed a competitive AUROC of 0.75 [95% CI, 0.74–0.76] for any AKI prediction, 0.82 [95% CI, 0.78–0.86] for predicting moderate-to-severe AKI, and 0.85 [95% CI, 0.78–0.89] for severe AKI prediction in 48 h. The model maintained a relatively robust performance when evaluated on samples stratified by age group

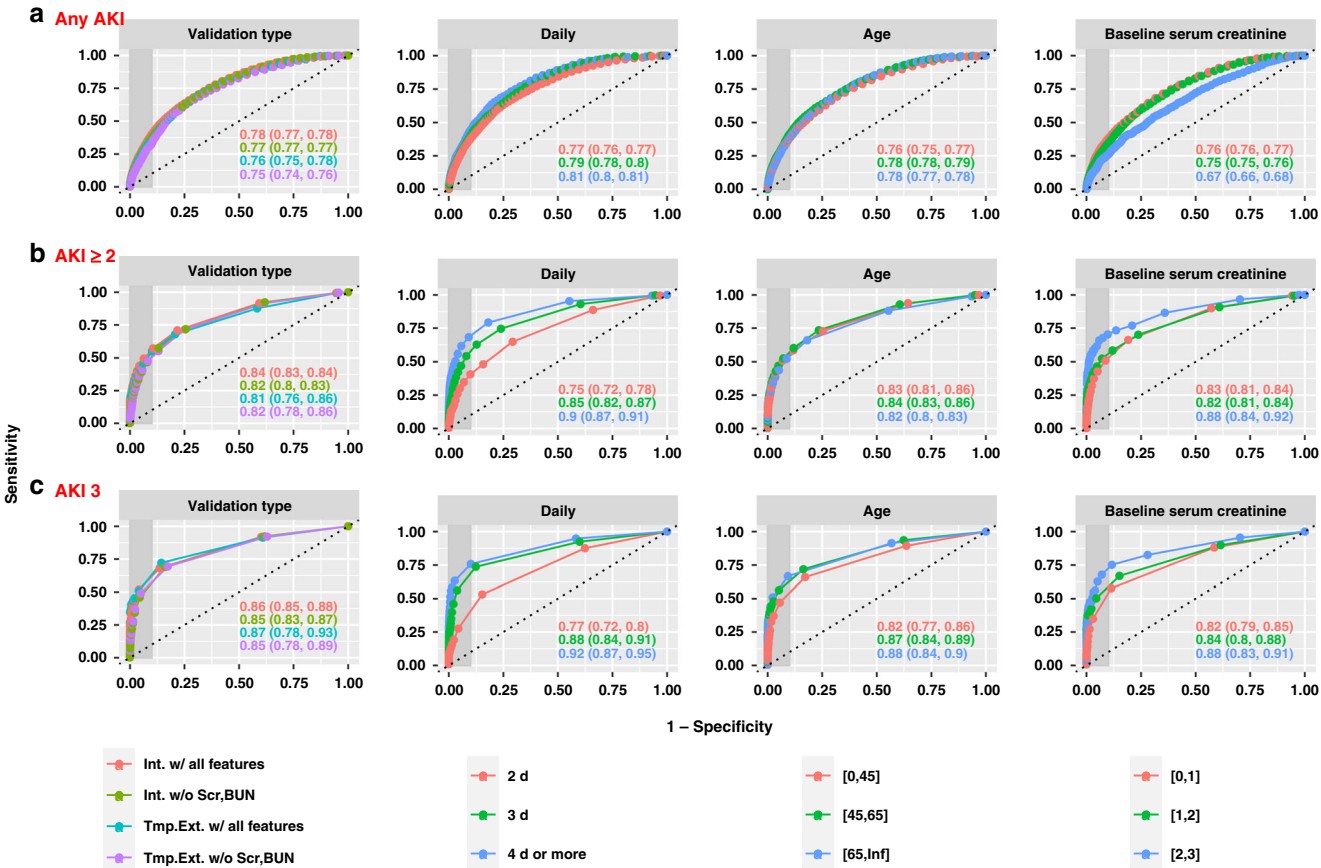

**Fig. 1 Model performance on the source health system data.** Model performance on the source health system data illustrated by receiver operating characteristic curves for predicting AKI events of any severity (**a**), at least stage 2 (**b**), or stage 3 (**c**) within the next 48-hours for various subgroups. "Int. w/ all features" refers to internal validation of models, i.e., random train/test sample split, using all extracted EHR features; "Int. w/o Scr, BUN" refers to internal validation of models that excludes SCr and BUN in all forms; "Tmp. Ext. w/ all features" refers to temporal validation of models, i.e., temporal hold-out set, using all extracted EHR features; "Tmp. Ext. w/o Scr, BUN" refers to temporal validation of models excluding SCr and BUN in all forms.

and baseline SCr. The model demonstrated improving performance over time and was steadily accurate after day 3 since admission, that is using data on and before day 4 to predict AKI in the next 2 days resulted in better accuracy than using data on and before day 2 to predict AKI in the next 2 days. The model for any AKI prediction showed calibration with a Hosmer-Lemshow score (HLS) of 42.8 ($p = 0.01$) for the temporal hold-out set, while it can be significantly improved to an HLS < 30 ($p > 0.02$) after simple recalibration (Supplementary Fig. 3). In Supplementary Figs. 4–6, we also reported the model performance for predicting AKI risk in 24 h, which showed uniformly better performance than 48-h prediction with temporal hold-out AUROC of 0.84 [95% CI, 0.82–0.84], 0.90 [95% CI, 0.88–0.93] for any AKI and moderate-to-severe AKI, and 0.93 [95% CI, 0.89–0.96] for severe AKI respectively. We also conducted experiments with the least absolute shrinkage and selection operator (*LASSO*) model, which showed significantly inferior performance compared with *DS-GBT*. The temporal validated AUROC and AUPRC for 48-h prediction of moderate-to-severe AKI using all features were 0.78 [95% CI, 0.73–0.84] and 0.06 [95% CI, 0.04–0.11] (Supplementary Fig. 7).

**Clinical interpretation of the AKI prediction model in source health system.** Using bootstrapped Shapely Additive exPlanations (SHAP) values, we looked inside the AKI prediction model by evaluating the marginal effects of predictive features identified by the model. We created a dashboard demonstrating the full list of feature importance ranking and marginal effect plots derived from our model (https://sxinger.shinyapps.io/AKI_shap_dashbd/). Here, we illustrate the marginal effects of top 10 important features from the *DS-GBT* model in predicting moderate-to-severe (clinical significant)[19] AKI in 48 h using all features (Fig. 2). The model identified SCr (*Creat SerPL-mCnc (2160-0)*) and its change, vancomycin exposure (*Vancomycin Injectable Solution_cumulative*), minimal value (*BP_SYSTOLIC_min, BP_DIASTOLIC_min*) and the hourly change of blood pressure (*BP_SYSTOLIC_slope, BP_DIASTOLIC_slope*), age, body mass index (*BMI*), height (*HT*), chest X-ray procedure (*X-ray of chest, frontal view*) as the top predictors of AKI. In the model without SCr and BUN (Supplementary Fig. 8), predictors identified emphasized on vancomycin exposure, blood pressure change, age, BMI, height, chest x-ray, and also identified piperacillin-tazobactam injection (*J2543: Injection, piperacillin sodium/tazobactam sodium*), bilirubin (*Billirub SerPI-mCnc (1975-2)*), and anion gap (*Anion Gap SerPI-sCnc (33037-3)*) as top 10 important factors.

Among the most important predictors identified from our study cohort, SCr and its change, BMI, and presence of a chest X-ray showed positive non-linear relationship with AKI risk. In particular, elevated SCr of more than 0.5 mg/dL increased the logarithmic odds ratio of AKI stage 2 or 3 in 48 h by 4 (equivalent to an increase of odds by $exp(4) = 54$ fold); exposure to vancomycin showed a persistently increased odds by over $exp(0.6) = 1.8$ fold; BMI above 40 increased odds by $exp(0.25) = 1.3$ fold; a anion gap above 13 mEq/L increased odds by $exp(0.25) = 1.3$ fold. Age, minimal blood pressure, and blood pressure changes showed U-shape associations with

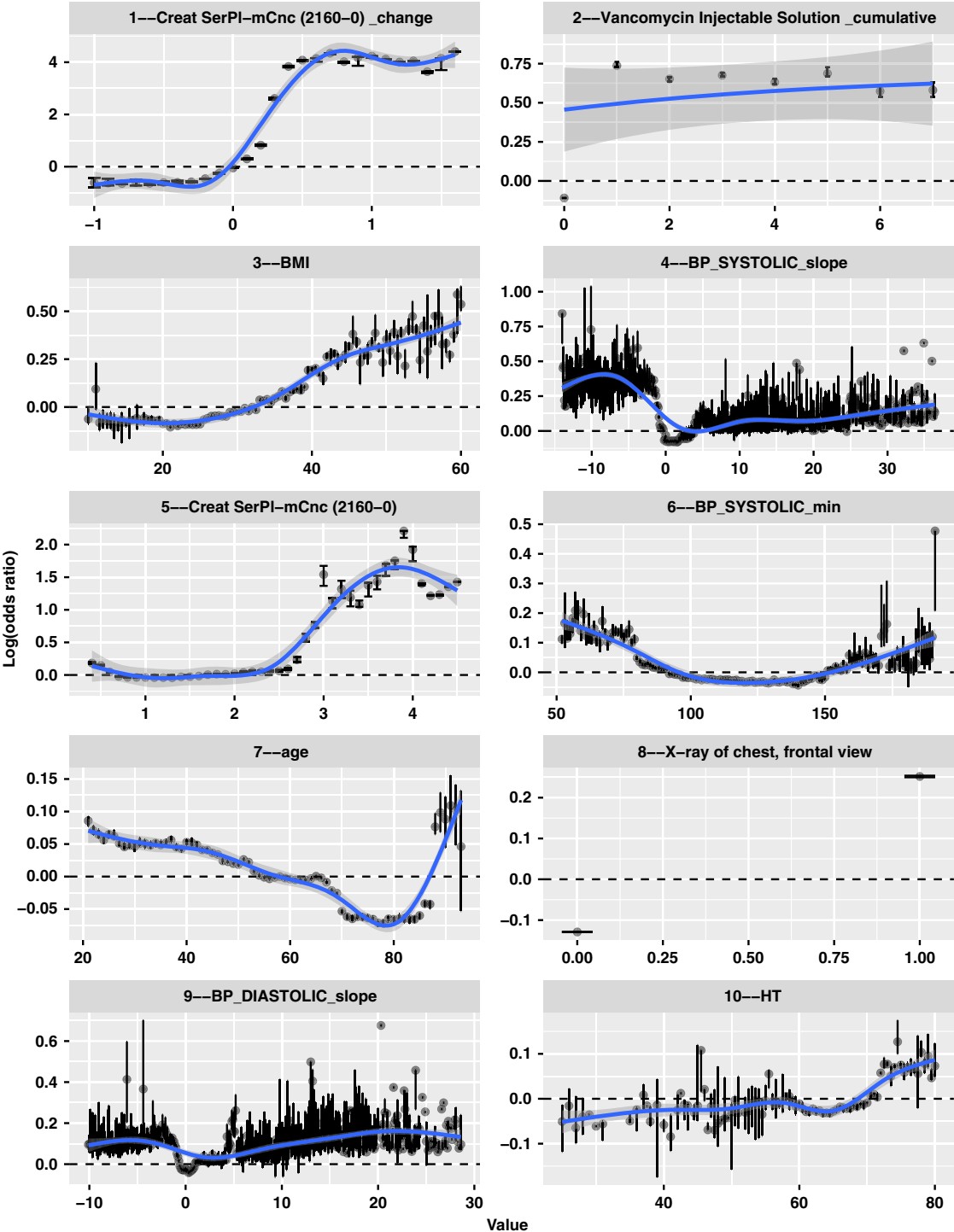

**Fig. 2 Marginal plots of the top 10 important variables for predicting moderate to severe AKI (at least AKI stage 2) in 48 h.** Each panel demonstrates marginal effects of each of the most impactful features ranked among the top 10 for predicting moderate-to-severe AKI in 48 h with SCr and BUN in the model. The x-axis gives the raw values of each feature and the y-axis gives the logarithmic of estimate odds ratio (i.e., the SHAP value) for moderate-severe AKI in 48 h when a feature takes certain value. Each dot represents the average SHAP value over bootstrapped samples with the error bars suggesting a 95% bootstrapped interval. The 'shaded area' represents the 95% confidence band for the LOWESS smoother extrapolating across all the dots. The full interactive dashboard can be found at: https://sxinger.shinyapps.io/AKI_shap_dashbd/.

AKI ≥ 2. Chest X-ray procedure was associated with an increased odds of $exp(0.3) = 1.3$ fold, while piperacillin-tazobactam injection increased odds by $exp(0.5) = 1.6$ fold. Age younger than 40 or older than 85 were identified to increase odds by $exp(0.05) = 1.1$ fold (detailed explanation is available in discussion); a minimal systolic blood pressure <75 mmHg or >150 mmHg both suggested an increase of odds by $exp(0.1) = 1.2$ fold; while a systolic blood pressure drop by more than 5 mmHg/hour were shown to be associated with increasing odds by at least $exp(0.4) = 1.5$ fold. At the patient-encounter level, we designed another dashboard (https://sxinger.shinyapps.io/AKI_ishap_dashbd/) to illuminate individualized predictors of moderate-to-severe AKI risk by decomposing the

**Table 1 Demographic characteristics at different health systems.**

| Demographic Characteristics | Site1 (reference) | Site2 | Site3 | Site4 | Site5 | Site6 |
|---|---|---|---|---|---|---|
| N | 153,821 | 100,819 | 86,264 | 57,286 | 19,542 | 88,865 |
| **Age** | | | | | | |
| 18–25 | 7819 (5.1) | 4879 (4.8) | 4223 (4.9) | 3193 (5.6) | 1172 (6) | 1978 (2.2)*** |
| 26–35 | 12,908 (8.4) | 8344 (8.3) | 7438 (8.6) | 4819 (8.4) | 2,007 (10.3) | 3218 (3.6)*** |
| 36–45 | 15,533 (10.1) | 10,401 (10.3) | 8315 (9.6) | 6150 (10.7) | 2132 (10.9) | 4724 (5.3)*** |
| 46–55 | 26,104 (17) | 15,658 (15.5)** | 13,566 (15.7)* | 10,517 (18.4) | 4184 (21.4)*** | 9885 (11.1)*** |
| 56–65 | 36,433 (23.7) | 22,373 (22.2)** | 18,768 (21.8)*** | 13,077 (22.8) | 5940 (30.4)*** | 15,564 (17.5)*** |
| ≥66 | 55,024 (35.8) | 39,164 (38.8)*** | 33,954 (39.4)*** | 19,533 (34.1)** | 4107 (21)*** | 53,496 (60.2)*** |
| **Sex** | | | | | | |
| Female | 75,962 (49.4) | 52,885 (52.5)*** | 42,894 (49.7) | 29,556 (51.6)*** | 8655 (44.3)*** | 43,748 (49.2) |
| Male | 77,857 (50.6) | 47,934 (47.5)*** | 43,370 (50.3) | 27,730 (48.4)*** | 10,887 (55.7)*** | 45,117 (50.8) |
| **Race** | | | | | | |
| White | 114,312 (74.3) | 64,839 (64.3)*** | 70,741 (82)*** | 50,499 (88.1)*** | 14,505 (74.2) | 81,282 (91.5)*** |
| Black | 24,046 (15.6) | 21,195 (21)*** | 9014 (10.4)*** | 5069 (8.8)*** | 3578 (18.3)** | 362 (0.4)*** |
| Asian | 1,416 (0.9) | 1,927 (1.9) | 711 (0.8) | 247 (0.4) | 299 (1.5) | 460 (0.5) |
| Native American | 594 (0.4) | 198 (0.2) | 863 (1) | 0 (0) | 75 (0.4) | 817 (0.9) |
| Other | 13,338 (8.7) | 5200 (5.2)*** | 4716 (5.5)*** | 294 (0.5)*** | 998 (5.1)*** | 433 (0.5)*** |
| Unknown | 115 (0.1) | 7460 (7.4)*** | 219 (0.3) | 1180 (2.1)** | 87 (0.4) | 5511 (6.2)*** |
| **Hispanic** | | | | | | |
| Yes | 8531 (5.5) | 11,636 (11.5)*** | 4284 (5) | 227 (0.4)*** | 889 (4.5) | 1001 (1.1)*** |
| No | 144,262 (93.8) | 84,958 (84.3)*** | 81,748 (94.8)*** | 0 (0)*** | 18,546 (94.9)*** | 81,789 (92)*** |
| Unknown | 1028 (0.7) | 4225 (4.2)*** | 232 (0.3) | 57,062 (99.6)*** | 107 (0.5) | 6075 (6.8)*** |
| **AKI stages** | | | | | | |
| Non-AKI | 129,230 (84.0) | 85,644 (84.9)*** | 73,172 (84.8)*** | 51,513 (89.9)*** | 16,579 (84.8)** | 76,370 (85.9)*** |
| Any AKI | 23,267 (16.0) | 15,175 (15.1)*** | 13,092 (15.2)*** | 5773 (10.1)*** | 2963 (15.2)** | 12,495 (14.1)*** |
| AKI Stage ≥2 | 3216 (2.1) | 2973 (2.9)*** | 2551 (3.0)*** | 1173 (2.0) | 620 (3.2)*** | 1416 (1.6)*** |
| AKI Stage 3 | 1562 (1.0) | 1655 (1.6)*** | 1441 (1.7)*** | 644 (1.1)* | 366 (1.9)*** | 577 (0.6)*** |

*p-value between 0.01 and 0.05; **p-value between 0.001 and 0.01; ***p-value < 0.001. All p-values were generated using Chi-square tests (two-sided).

predicted logarithmic odds ratio into a summation of SHAP values, where each value represents the individualized effect of a risk factor.

**External validation of the AKI prediction model in target health systems.** We conducted external validation of the above AKI prediction model built from a single health system on five other health systems. Table 1 shows varied demographic profiles across different healthcare systems (i.e., sites). With Site1 being the reference source health system, Site4 showed a similar age decomposition, while patients at Site2, Site3, and Site6 were significantly older. Gender distribution of Site3 and Site6 were similar to the reference site, while Site2 and Site4 biased towards females and Site5 towards males. Although the dominating race was Caucasian, the proportion varied significantly across sites. Site2 and Site5 had significantly larger groups of African Americans and Asians respectively, while Site3 had a significantly higher proportion of Native Americans. In addition, Site2 had a significantly larger Hispanic population, while Site4 seemed to not document the ethnicity information well. Furthermore, the 5 external validation sites had significantly lower rates of any AKI, but Site2, Site3 and Site5 had significantly higher prevalence rates of moderate-to-severe AKI (AKI stage 2 and 3).

Figure 3 shows the external validation performance in terms of AUROC for predicting different stages of AKI in the next 48 h using all variables versus model without SCr and BUN in any forms. We compared the performance of direct transportation of the AKI prediction model trained from the source site to target sites without any adaptation (i.e., Transported Model) against refitting the model on target site data (i.e., Refitted Model). External validation performance of the transported model showed large variations across sites with AUROC for predicting moderate-to-severe AKI (i.e., at least stage 2) ranging from 0.68 [95% CI, 0.66–0.71] to 0.80 [95% CI, 0.77–0.82], and AUPRC ranging from 0.09 [95% CI, 0.07–0.12] to 0.15 [95%

CI, 0.12–0.19] (Supplementary Fig. 9). We observed significant AUROC gain from refitting the same model using local data at target sites. For example, for any AKI prediction, refitted model at Site5 achieved an AUROC of 0.83 [95% CI, 0.81–0.85] (AUPRC of 0.15 [95% CI, 0.13–0.17]), comparing to 0.71 [95% CI, 0.70–0.71] (AUPRC of 0.08 [95% CI, 0.08–0.09]) using the transported model from the source system Site1. F-test between Hosmer-Lemeshow Chi-square statistics suggested that calibration was more comparable for moderate-to-severe AKI (Supplementary Fig. 10). For example, 3 out of 5 sites did not incur significant calibration improvement ($p > 0.1$) for AKI-stage 3 prediction using all features; and 3 out of 5 sites did not show significant calibration deterioration for predicting moderate-to-severe AKI without SCr and BUN (Supplementary Fig. 10). Site5 was well-adjusted to the static model because the transported model was not significantly worse-off than the refitted one for predictions of all AKI stages (Supplementary Fig. 10). In Supplementary Figs. 11–13, we also reported the external validation results of models for predicting AKI risk in 24 h, which showed uniformly better performance than 48-h prediction, with transported and fitted AUROC ranging from 0.66 [95% CI, 0.65–0.67] to 0.81 [95% CI, 0.81–0.82] and from 0.83 [95% CI, 0.82–0.83] to 0.89 [95% CI, 0.88–0.89] respectively, for any AKI prediction. We also externally validated the *LASSO* model, which significantly underperformed the *DS-GBT* model, with AUROC for predicting moderate-to-severe AKI in 48 h (using all features) ranging from 0.68 [95% CI, 0.66–0.69] to 0.71 [95% CI, 0.70–0.73], and AUPRC ranging from 0.02 [95% CI, 0.02–0.03] to 0.1 [95% CI, 0.08–0.1] (Supplementary Fig. 14).

**Source of performance heterogeneity.** To better understand the performance superiority of the refitted models over the transported model, we performed a qualitative analysis by looking into the most important predictors selected by the refitted models at

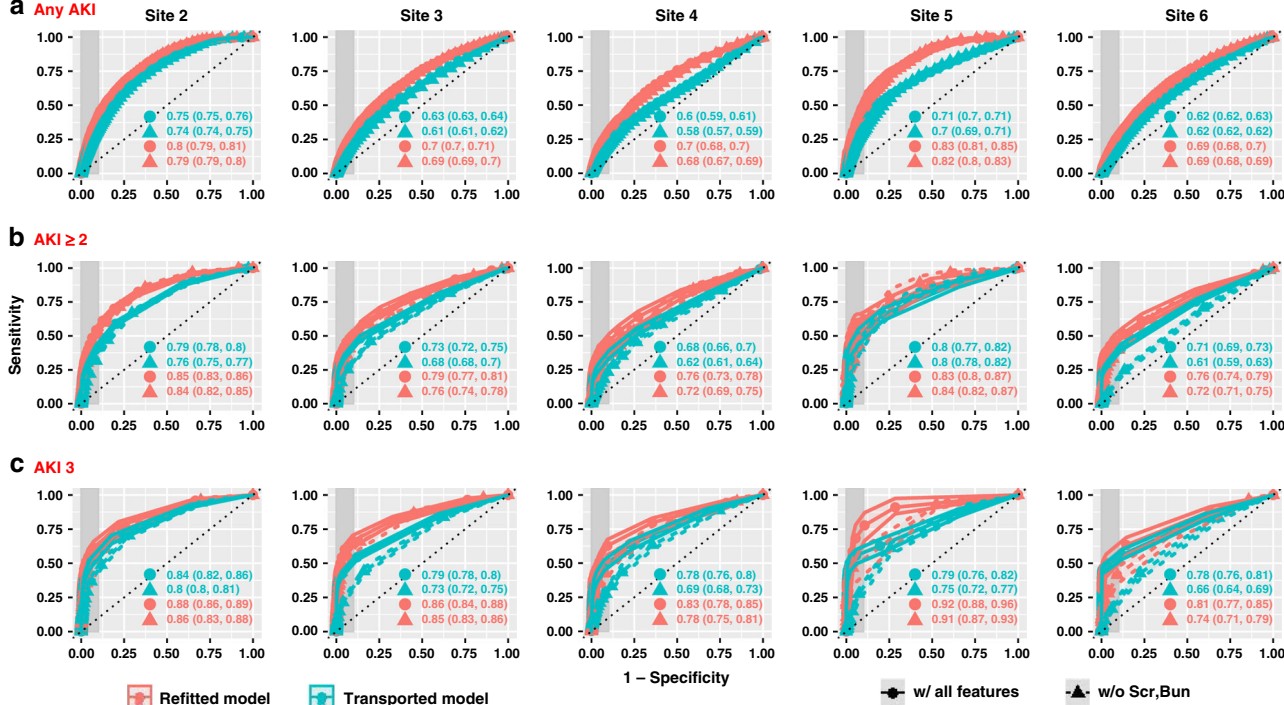

**Fig. 3 External validation of the AKI prediction model. a–c** Comparison of model performance illustrated by receiver operating characteristic curves for transported model vs refitted model in 48-h AKI predictions on external validation site data.

the external sites. Figure 4 shows how common vs specific features were in relation to their importance rankings in the predictive models, with the y-axis being the percentage of sites that selected certain features (i.e., 1.0 = the most common feature selected by all refitted models, 0.17 (1/6) = the most specific feature deemed important by only one site) and x-axis showing the importance feature ranking averaged across sites (i.e., the closer it is to 1, the more important the feature is).

There are many more site-specific and important features, especially from medication and lab test categories. For example, cumulative exposure of *Vancomycin Injectable Solution* and *Piperacillin 4000 MG / tazobactam 500 MG Injection* were two medication representations that were highly predictive of AKI but were only specific to one site. A group of glucose-related labs, i.e., *Glucose Blood Manual Strip*, *Glucose Blood Strip Auto* and *Glucose Serum* were shown to be predictive but only specific to one or two sites. Other labs, such as *Osmolality in Serum* and *Anion Gap3 in Serum*, were also highly predictive for only one site; $CO_2$ *in Serum*, *Vancomycin Trough in Serum*, *Albumin in Serum by Bromocresol green (BCG) dye binding method*, and *Prothrombin time* were predictive at two sites. The most common and predictive features across all six sites were *Serum Creatinine*, *Height*, *BMI*, *Age*, *INR PPP* and *Hgb in Blood*, while the second most common and predictive features include a variety of blood pressure summaries, *Serum Bilirubin*, *Serum Chloride*, *Serum Potassium*, and *Serum Phosphate*.

Among the common set of important features, we analyzed their marginal effects on the likelihood of AKI and observed variations. SHAP value extrapolations of the top 4 commonly important variables in predicting at least AKI stage 2 within 48 h across all 6 sites are illustrated in Supplementary Fig. 15. For the model with all features (15a.), age, BMI, SCr and SCr changes showed relative consistent associations with AKI risks. However, there were noticeable variations on the strength of such associations across sites. For example, Site3 data showed no significant relationship between age and log odds of moderate-to-severe AKI events, while all the other

sites suggested non-positive associations, which means that the moderate-to-severe AKI risks among younger hospitalized patients are relatively higher than the older population. On the contrary, the effect of age is showing a positive association with any AKI prediction in 48 h (Supplementary Figs. 16 and 17). For the model excluding SCr and BUN (15b.), there was a major disagreement in the marginal effects of blood hemoglobin (LOINC:718-7). Site1 data demonstrated a negative association between blood hemoglobin and AKI risk, Site4 suggested the opposite, while the remaining sites did not identify significant effects from blood hemoglobin.

**A metric for evaluating model transportability**. Prediction performance discrepancy between the transported and refitted models may be caused by population and feature space differences, however very few studies measured the feature space differences to estimate potential performance variation when models are transported from one hospital to another. Here, we developed a metric measuring the joint distribution of features space, coined as the adjusted maximum mean discrepancy (adjMMD), to explain the model performance variabilities (See details in "Methods"). adjMMD is a modified version of the classic MMD metric, which has been a widely adopted test statistic for measuring data distribution shift between source and target data in transfer learning[20]. Supplementary Figs. 20 and 21 demonstrate the anticipated properties of adjMMD with respect to the number of top-ranking variables: (a) the metric increased significantly when an important variable is completely missing (e.g., the second most important variable in our source model was a specific RxNorm code for *Vancomycin Injectable Solution*, which was completely missing from all external validation sites due to differences in medication representation in the RxNorm hierarchy, that accounted for the sudden jump of adjMMD); (b) the metric recovered, reflected by the gradual decreasing of adjMMD, when better distribution matching occurred after additional variables were included (e.g., the third most important variable, BMI, was

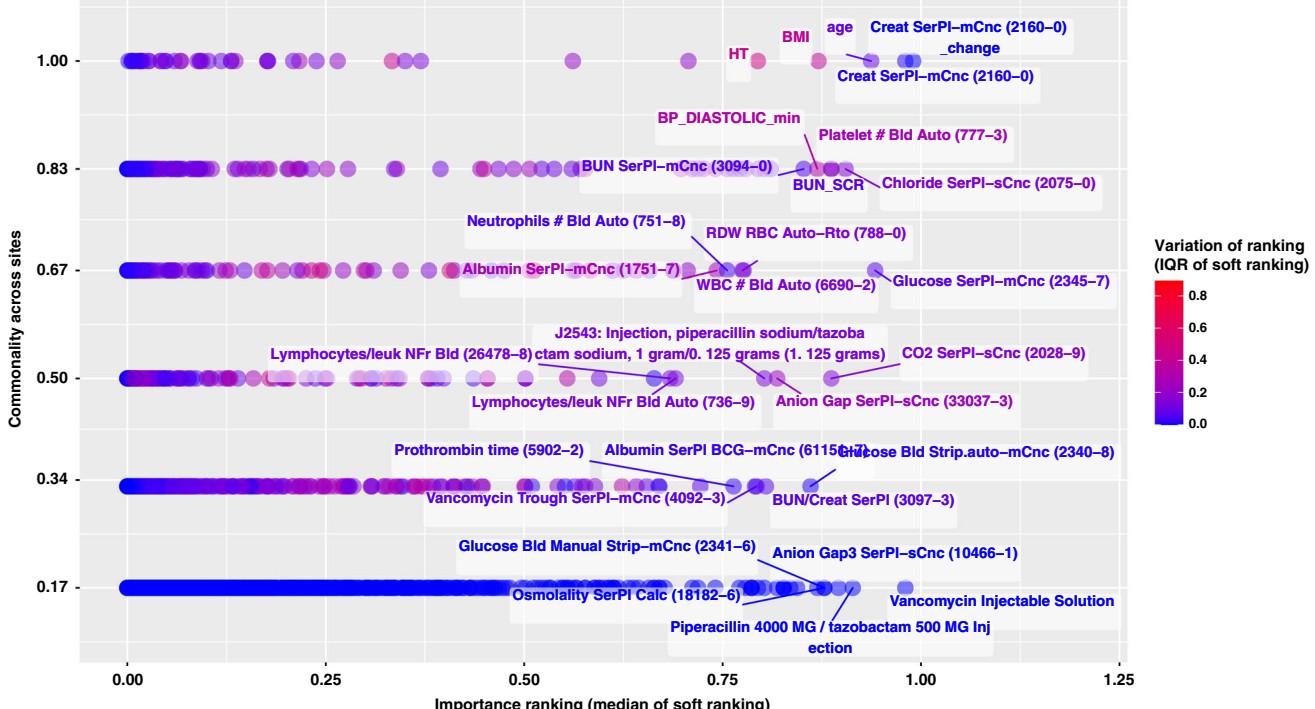

**Fig. 4 Feature selection disparities across sites (48-h predictions for at least AKI stage 2).** The figure demonstrates feature importance disparities for the models trained on source data as well as refitted models at each validation site, using all features. Each dot corresponds to one of the most important features ranked among the top-100 by at least one of the six models; *y*-axis measures the proportions of sites that identified the feature as top-100, or "commonality across sites"; *x*-axis measures the median of variable importance rankings measured as "soft ranking" (the closer it is to 1, the higher the feature ranks), which is also color coded by the interquartile range (IQR) of the ranks across sites (the higher the IQR is, the more disagreement across sites on the importance of that feature). Top-100 is an arbitrary cutoff we used to analyze the most important features to illustrate heterogeneity. We also reported similar feature disparity figures for models predicting moderate-to-severe AKI prediction without SCr and BUN (Supplementary Fig. 18), as well as models predicting any AKI (Supplementary Figs. 16 and 19).

well populated in all of the validation sites with similar distributions as the source site).

More importantly, we demonstrate that adjMMD can be used to infer model performance deteriorations (i.e., drop of AUROC or ΔAUC). In order to improve the practicality of adjMMD such that target sites would not need to collect all variables to make a quick inference about model transportability, we further developed a simple procedure to identify a sufficiently small feature set (*minimal feature set*) required to accurately infer the AUROC variation based on Pearson correlation coefficient and regression. Figure 5 shows the results of this procedure for models predicting moderate-to-severe AKI using all features. Figure 5a shows that the Pearson correlation coefficient between adjMMD and ΔAUC rose drastically to 0.95 when top 13 variables were included, gradually dropping afterwards and converging slowly to 0.91 when more variables were considered. In other words, by calculating adjMMD using only the top-13 important variables selected by the source model against the target hospital data, we can accurately predict the performance change on data from any target hospitals. The performance change can be simply calculated using the simple linear equation $y = -0.018 + 0.344x$ (Fig. 5b.), with *x* denoting adjMMD and *y* the ΔAUC. It indicates that every 0.1 unit increase in the adjMMD would potentially lead to 0.0344 decrease in the target AUROC, or equivalently, an expected AUROC of 0.77 ($= 0.81 - 0.344 \times 0.1$) when transporting the established KUMC model to a new site for 48-h prediction of moderate-to-severe AKI. For the model in predicting moderate-to-severe AKI with SCr and BUN removed, we applied the same procedure and identified a sufficiently small set of 33 features for predicting the transportability of the corresponding

model with the equation $y = -0.028 + 0.367x$ (Supplementary Fig. 22).

We further tested the robustness of adjMMD with the following three sets of experiments for 48-h prediction of moderate-to-severe AKI: (1) using each one of the other six sites as derivation site and iteratively evaluating the correlations between adjMMD and ΔAUC (6 models); (2) conducted the same experiments using LASSO model (6 models); (3) conducted the same experiments with a limited version of GBT (*limited-GBT*) model using only the common set features (6 models). Panel A. of Supplementary Fig. 23 depicts the Pearson correlations between adjMMD and ΔAUC with increasing number of important variables included. It showed similar trend as in Fig. 5a, that is: (a) the more variables included, the stronger correlation between adjMMD and ΔAUC which eventually converged. Among the 18 models, 16 achieved a Pearson correlation above 0.9. (b) There exists a minimal feature set that can achieve optimal or sub-optimal Pearson correlation between adjMMD and ΔAUC for each model. However, minimal feature set size may vary, leading to variant slopes of the fitted regression lines between expected ΔAUC and adjMMD as demonstrated in Panel B. Nonetheless, all linear associations were significantly positive, confirmed by fitting a mixed-effect model: ΔAUC, $y = -0.014 + 0.257x$ with a 95% CI for the slope to be [0.08, 0.44]. This mixed-effect model accounted for random effects from model and derivation site difference (more details are described in Method). We also performed two validation tests: (a) *sample-agnostic tests*, where we derived the linear equation from the same derivation site (*same site/sample*) and evaluated how well it fits to the other sites (*different site/sample*); and (b) *model-agnostic tests*, where we

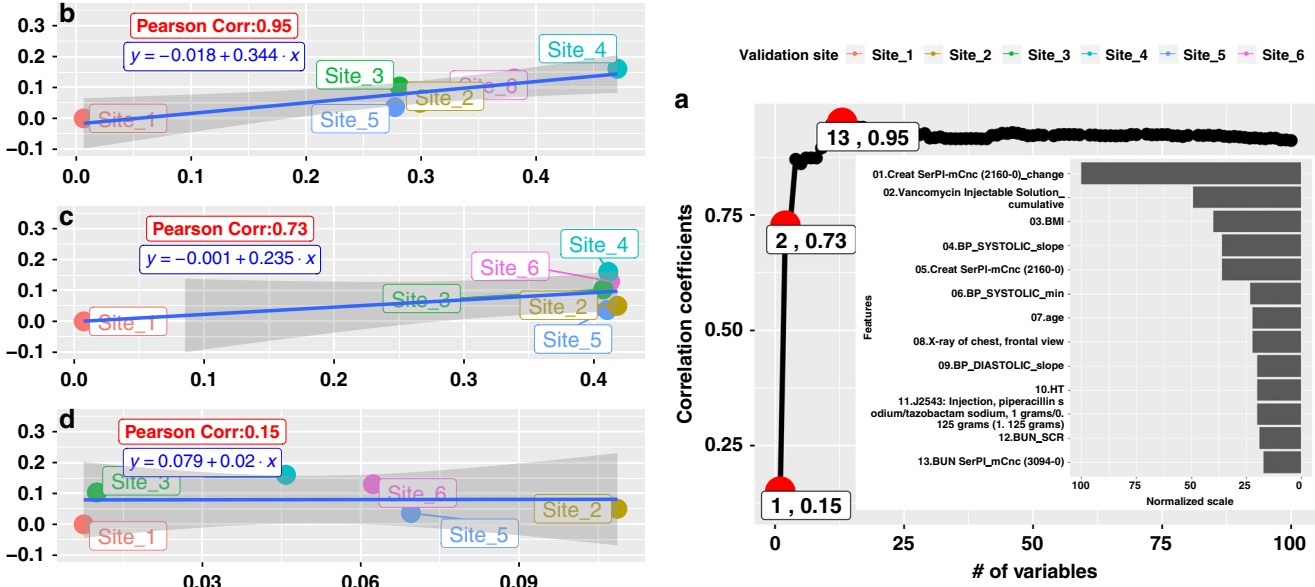

**Fig. 5 Correlation Between adjMMD and Prediction Performance (AUC) for predicting moderate-to-severe AKI in 48 h).** It shows full details on how adjMMD can be used to infer the prediction performance drop. Panel **a** on the right illustrates the Pearson Correlation Coefficient between adjMMD value and AUC drop as a function of the number of variables included (e.g., the first red circle with a label of "1, 0.15" suggests that if only the most important variable is considered, the correlation between adjMMD and AUC drop is 0.15). **b**, **c**, **d** on the left demonstrate the simple regression lines between adjMMD and AUC drop with 95% confidence band shaded. Take d as an example, when only the most important variables (*Create SerPL-mCnc (2160-0) change*, or *SCr change*) are considered, the adjMMD is barely predictive of ΔAUC. The more features are included in the adjMMD measure, the better it is at anticipating AUROC deterioration. Experimental results suggest that when top 13 important features are included for predicting moderate-to-severe AKI in 48 h, the strength of association between adjMMD and ΔAUC reaches an optimal value of 0.95 measured by Pearson correlation coefficient.

derived the linear equation from the same model (*same model*) and evaluated how well it fits to the other models (*different model*). Supplementary Fig. 24 showed that there were no significant differences in residuals sum of square (RSS) between each comparison groups, with $p$-value $= 0.66$ for the *sample-agnostic test* and $p$-value $= 0.57$ for the *model-agnostic test*.

## Discussion

The DS-GBT model for predicting the development of AKI across all hospitalized patients with a lead time of 48 h was developed and validated using the PCORnet infrastructure and leveraged all key data elements currently available and well populated in the CDM tables, which required minimal effort in transporting the model to other PCORnet sites (PCORnet currently represents more than 68 million patients in the United States). With internal and temporal validated AUROC of 0.84 and 0.81 respectively for predicting moderate-to-severe AKI (0.78 and 0.76 for any AKI), as well as an externally validated AUROC of 0.7 for the combined dataset of all external sites and 0.8 for the best site (0.69 for the combined set and 0.75 for the best external site on any AKI prediction), the experiments have shown the validity of our prediction model. Compared to the state-of-the art work by Koyner et al.[8,9], our model achieved better performance for any AKI prediction and competitive results for moderate-to-severe AKI prediction, but with much-reduced data pre-selection and processing effort as a result of CDM utilization. It is still unclear to us how the Google DeepMind model[10] would perform on a more general hospitalized population. Our focus here is to develop explainable or "white-box" AKI prediction models that can be efficiently transported across hospitals.

Clinical interpretability of machine-learning models is utmost important in clinical practice because it may be difficult or impossible to detect subtle shortcomings of the accuracy-driven black-box models[21]. We have improved transparency of the

model by demonstrating both the feature importance ranking and SHAP-value-based marginal plots via an interactive dashboard, which enables knowledge discovery. It is worth noting that the majority of the marginal effects derived from our models are nonlinear, and some are not even monotonic. In addition to the known positive associations with SCr, BMI and vancomycin exposure identified by the model, we observed many non-conventional positive associations. The positive effect of chest X-ray procedure on AKI risk may be a latent indicator of patient's overall disease severity, as it is shown to be highly correlated with ICU admission (OR = 3.4, $p$-value $< 0.001$), intubation (OR = 3.0, $p$-value $< 0.001$), and pneumonia diagnosis (OR = 2.7, $p$-value $< 0.001$). The non-positive association identified between age and moderate-to-severe AKI and the positive monotonic relationship between age and any AKI may imply that younger patients (more specifically, younger than 55 years old) who are staying at the hospital for two days or more were at higher risk for more severe kidney injury. This relationship is also in line with the univariate U-shape associations between age and AKI stage across health systems (Supplementary Fig. 25), such that the AKI stage 1 patients are consistently older than patients developing AKI stages 2 and 3. When examining the distributions of patients' diagnoses present on admission (POA), we observed different patterns of reasons for admission among younger patients developing AKI 1 and AKI ≥ 2. Among the top 10 reasons for admission among patients younger than 35, 45 or 55 years old, there were a disproportionally higher number of patients admitted for *other aftercares* (usually due to auto injuries), *cardiac dysrhythmias*, *hypertension with complications*, *other nervous system disorders*, *diseases of white blood cells*, and *other nutritional/endocrine/metabolic disorders*, among AKI ≥ 2 than AKI stage 1 or non-AKI (Supplementary Fig. 26). Some of these conditions (e.g., *diseases of white blood cells*, and *other metabolic disorders*) may suggest auto-immune spike among younger

patients who are more likely to have intrinsic AKI. The U-shape association between minimal blood pressure (*BP_Systolic_min*) and blood pressure change per hour (*BP_systolic_slope*, *BP_diastolic_slope*) suggests two subtypes of mechanisms relating blood pressure to AKI risk: both hypertension and hypotension[22] (or drastic decrease of blood pressure) are risk factors of AKI.

The PCORnet CDM infrastructure offers an important platform for testing model transportability at a lower cost. Experiments showed that the refitted models on local data generally resulted in better accuracy than directly applying the transported model. It draws our attention on a data transformation problem that there is a lack of consensus standard for mapping local concepts to common terminology. The CDM may have standardized the language for representing clinical elements, but not necessarily harmonized the vocabulary used in different clinical settings. For example, there are two types of glucose labs identified to be important at different sites, one is serum glucose (LOINC: 2345-7) and the other is glucose in blood by automated test strip (LOINC: 2340-8). As discussed in our previous work, one needs to be cognizant that clinical data may be encoded at different levels of granularities such as National Drug Codes for specific brands and formulations versus high-level representation of ingredient or routed drug available in RxNorm[23]. In this study, we further confirmed the importance of representing medical concepts at appropriate granularity level. Take *vancomycin* as an example, all six sites used different sets of RxNorm (at different granularity level) to encode *vancomycin injectable solution*, while only the abstraction used at the training site seemed to generate a signal strong enough to be recognized by the model. Demystifying the prediction models not only helped us understand the models better, but also enabled further diagnoses of the source of heterogeneity besides the case-mix and demographic differences that were more straightforward to understand. For example, as shown in Supplementary Fig. 27, we observed that the association between hemoglobin in blood (*Hgb Bld-mCnc (718-7)*) and moderate-to-severe AKI risk were opposite between Site1 (positive) and Site4 (negative). As we further investigated, we discovered that this opposite association was potentially caused by the fact that the hemoglobin levels recorded at Site4 were systematically higher than Site1 among moderate-to-severe AKI cases, while there was no significant distribution difference for AKI stage 1 cases.

Our experimental results demonstrated that performance deterioration of the transported model is likely due to differences in population factors and differences in data vocabulary and documentation. Building a truly generalizable model requires truly representative data, which is almost infeasible with healthcare data siloed at individual hospitals and sharing of data hampered by privacy concerns. If performance variation is inevitable, then we need to be prepared and proactive. A solution we proposed in this study is adjMMD, which can be used as the precursor for inferring the effectiveness of a prediction model when transported to other hospitals. This metric not only provides statistical evidence for deciding whether direct model transportation would be sufficient, the dynamic of adjMMD also offers a path to discover potential sources of data heterogeneity contributing to model performance drop. We envision that adjMMD could be used as an informative tool to accelerate the external validation process and adaptation of external machine-learning models. The sample- and model-agnostic tests demonstrated that adjMMD can be used as a robust indicator of performance change over varying training sets, models, and feature spaces. From an implementation perspective, with multi-center testing data, one can develop such an auxiliary linear equation between adjMMD and ΔAUC by following the experimental procedures described in this study to inform the transportability

of any established model to future sites. Hospitals across the US and globally have different data/informatics infrastructure maturity and AI capability[24]. When a prediction model is transported to a hospital with less mature clinical data warehouse (i.e., insufficient amount of quality data for building and validating an independent model), users can use the adjMMD score to decide if the externally trained model can be used in the clinic as is with acceptable performance. Alternatively, if the target hospital is mature in both data and AI capability, adjMMD can then be used to decide if the model needs to be updated with varying effort ranging from simple recalibration to model refitting on local data and even full model revision with the incorporation of new predictors.

It is anticipated that earlier identification of patients at high risk of developing AKI will be coupled to earlier interventions that might improve clinical outcomes. While there are currently no specific interventions that can prevent AKI, there are general measures such as optimizing fluid and hemodynamic status, avoiding nephrotoxic exposures, and delaying procedures such as surgery. There is some evidence that implementation of "care bundles" consisting of such measures, if implemented early (e.g., in response to an electronic alert), is associated with improved outcomes, including improved recovery from AKI, shorter length of stay, and/or reduced mortality[25–27].

There are several limitations to our study. First, the definition of AKI depended on SCr changes in reference to patients' baseline SCr, which is not always observable (2/3 of AKI are community acquired[28]). For patients without any SCr observed within 7 days prior to admission, we had to use their admission SCr as the baseline, which may underestimate the true AKI incidence. Second, since patients were censored at 7 days, our model was not validated in those who had a length of stay >7 days. Third, we used CPT billing codes to capture procedures performed during a hospital stay because PCORnet CDM currently does not collect procedure orders that are available in real time in the EMR. Fourth, although PCORnet CDM has evolved over time by incorporating more data elements, it is still missing many key variables such as heart rate, oxygen saturation, and Braden scale score, which have been shown to be important AKI risk factors[8]. Lastly, although patient and data heterogeneity are the reality, advanced data harmonization techniques may mitigate the problem. Other potential algorithmic solution to improve model transportability also include statistical model updating methods[14] and transfer learning, an emerging learning paradigm in machine learning.

## Methods

**Data source.** The clinical data assembled for this study were collected by the Greater Plain Collaborative (GPC), a Patient-Centered Outcome Research Network (PCORnet) Clinical Data Research Network (CDRN) including twelve healthcare systems in nine states[18]. PCORnet developed the CDM to support federated research networks by centering its schema on the patient entity, and enforcing data mapping to controlled vocabularies such as Current Procedural Terminology (CPT), SNOMED CT, Healthcare Common Procedure Coding System (HCPCS), the ICD versions 9 and 10, Logical Observation Identifiers Names and Codes (LOINC), and RxNorm[15]. Using the CDM at all GPC institutions/sites that pass quarterly data quality assessments allows efficient query and analysis execution across different instances of the data model[29]. For this study, six participating GPC sites included the University of Kansas Medical Center, University of Texas Southwestern, University of Nebraska Medical Center, University of Missouri, Medical College of Wisconsin, and Marshfield Clinic Research Institute. CDM data provided by each site were de-identified to meet the Health Insurance Portability and Accountability Act of 1996 (HIPAA) 'Safe Harbor' criteria. This study was determined not to be human subject research by the institutional review board of the GPC consortium because it only involved the collection of existing and de-identified patient medical data.

We developed a data extraction package specific to PCORnet CDM and extracted data comprising all patients aged between 18 and 90 who were hospitalized for at least 2 days with at least two SCr records from the beginning of 2010 to the end of 2018. Patients were excluded if they had evidence of severe

kidney dysfunction at or before admission, that is (a) estimated glomerular filtration rate <15 mL/min/1.73 m², or (b) has undergone any dialysis procedure or renal transplantation (RRT) prior to the visit, or (c) required RRT within 48 h of their first documented SCr measurement. Burn patients were also excluded since SCr becomes a less reliable tool in assessing renal function during hypermetabolic phase[30]. Patient inclusion and exclusion procedures are summarized in Supplementary Fig. 28.

**Data processing**. For each patient in the data set, we collected all variables that are currently populated in the PCORnet CDM schema over both source and target sites, which includes: general demographic details (i.e., age, gender, and race), all structured clinical variables that are currently supported by PCORnet CDM Version 4, including diagnoses (ICD-9 and ICD-10 codes), procedures (ICD and CPT codes), lab tests (LOINC codes), medications (RXNORM and NDC codes), as well as selected vital signs (e.g., blood pressure, height, weight, BMI)[29]. All variables are time-stamped and every patient in the dataset was represented by a sequence of clinical events construed by clinical observation vectors aggregated on daily basis (Extended Fig. 1), so that the feature set formed by data prior to or on day $t$ could be used to predict AKI within $[t + 1, t + 2]$ days for 48-h prediction (or within the next day for 24-h prediction).

The initial feature set contained more than 30,000 distinct features. We performed an automated curation process as follows: (1) systematically identified extreme values of numerical variables (e.g., lab test results and vital signs) that are beyond 1st and 99th percentile as outliers and removed them; (2) performed one-hot-coding on categorical variables (e.g., diagnosis and procedure codes) to convert them into binary representations; (3) used the *cumulative-exposure-days* of medications as predictors instead of a binary indicator of the sheer existence of that medication; (4) when repeated measurements presented within the certain time interval, we chose the most recent value; (5) when measurements are missing for a certain time interval, we performed a common sampling practice called *sample-and-hold* which carried the earlier available observation over; (6) introduced additional features such as *lab value changes since last observation* or *daily blood pressure trends*, which have been shown to be predictive of AKI[8].

**Experimental design**. For each of the six prediction tasks (24 or 48-h prediction of the 3 AKI stages), we repeated the following steps for model training, validating/transporting, model refitting, and post-analysis: (a) we first split both source and target data into derivation, calibration, internal validation and temporal validation sets, with calibration and internal validation being 15% of randomly held-out inpatient encounters occurred between 1 January 2010 and 31 December 2016 and temporal validation occurred after 1 January 2017; (b) at the training stage, we tuned a set of hyperparameters (learning rate, number of trees, depth of trees, sample rates, and number of cases in leave notes) with Bayesian optimization approach using fivefold cross validation and retrained a final reference model using all training data (source); (c) at the validation stage, we re-calibrated the model on the calibration source data set and then evaluated the model performance on both the internal and temporal validation sets within source data, as well as the external validation set within target data; (d) at the refitting stage, we iteratively repeated (a) through (c) with models developed on training data from each target site and validated on corresponding validation sets; (e) finally, at the post-analysis stage, we evaluated the adjMMD and formed a simple analytic set of adjMMD and ΔAUC (i.e. the difference between AUROC of internal temporal validation set and external validation sets), which were used to derive the simple linear regression equation to infer ΔAUC with observed adjMMD.

**AKI definition**. According to the Kidney Disease Improving Global Outcomes (KDIGO) clinical practice guideline for AKI, we staged AKI for severity based on the SCr-based criteria[31]. We did not use urine output to define AKI because it was not recorded in most of the sites in the GPC network and is less likely to be accurate outside the critical care environment. KDIGO accepts two SCr-based definitions of AKI of any severity ('any AKI'): (1) an increase in SCr of 0.3 mg/dL (26.5 μmol/L) within 48 h or (2) an increase in SCr of 1.5 times the baseline creatinine level of a patient, known or presumed to have occurred within the previous 7 days. This study defined baseline creatinine as the most recent SCr when previous measurements were available, or admission SCr value when past measurements were not available. Moderate AKI ('AKI stage 2') is defined as an increase in SCr of 2.0 to 2.9 times the baseline within the previous 7 days. The most severe AKI ('AKI stage 3') is defined as either an increase in SCr to 4 mg/dL (353.5 μmol/L) after an acute increase of at least 0.3 mg/dL within 48 h, or an increase in SCr of more than three times the baseline within the previous 7 days, or an initiation of renal replacement therapy.

To realize a discrete-time survival framework[32], the AKI stages were computed at times at which there was a SCr measurement present in the sequence, and then copied forward in time until the next SCr measurement, at which time the ground-truth AKI state was updated accordingly. Independent right-censoring was achieved by fixing a maximum follow-up period at $\tau$ (we chose $\tau = 7$ based on expert knowledge), to reduce the further impact on data imbalance resulting from the right skewedness of AKI onset times.

**Learning algorithm**. We chose Gradient Boosting Tree-based Machines (GBT) as the learning model and then combined it with a discrete-time survival framework using independent censoring. GBT is a family of machine-learning techniques that have shown considerable success in a wide range of practical applications. We chose GBT as the base learner not only for its robustness against high-dimensionality and collinearity, but also because it embeds feature selection scheme within the process of model development, making its output explainable. To better control overfitting, we tuned the hyper-parameters (depth of trees: 2–10; learning rate: 0.01–0.1; minimal child weight: 1–10; the number of trees is determined by early stopping, i.e., if the holdout area-under-receiver-operating-curve had not been improved for 100 rounds, then we stopped adding trees) within training set using 10-fold cross validations. It simulates a discrete-time survival framework by separating the full course of patient's stay history into $L$ non-overlapping daily windows, $L = 1\Delta t, 2\Delta t, \dots T$, where $T$ is pre-set to total length of stay or an arbitrary censor point (e.g., 7 days since admission) and $\Delta t$ can be chosen based on clinical needs (e.g., $\Delta t = 1$ is for 1-day (24 h), $\Delta t = 2$ is for or 2-day (48 h) prediction, etc.). We would use all available variables up to time $t - 1$ to predict AKI risk of various stages in $t$. Timestamped indicators were created to implicitly associate events that occur within the same time window.

Missing values were handled in the following fashion: for categorical data, a value of 0 was set for missing; while for numerical data, a missing value split was always accounted for and the "best" imputation value can be adaptively learned based on improvement in training AUROC, at each tree node within the ensemble. For example, if a variable $X$ takes values [0,1,2,3,NA, NA], where "NA" stands for missing, the following two decisions will be made automatically at each split for each tree: (1) should we split based on "missing or not"; (2) if we split based on values, e.g., >1 or ≤1, should we merge the missing cases with the bin of >1 or ≤1. However, if available values of certain variable exist in the earlier time window, they will be carried over for predictions in the future.

**Evaluations**. We used the area under the receiver operator curve (AUROC) and area under the precision-recall curve (AUPRC) to compare the overall prediction performance, with the latter known to be more robust to imbalanced datasets. We also validated our model over a variety of sub-cohorts, stratified by age group, by baseline SCr level, and by days since admission, on both source and target cohorts. In addition, we characterized calibration by Hosmer-Lemeshow (HL) Chi-squared score and used F scores to compare the HL scores between transported and refitted models. Feature importance was ranked based on "gain", the cumulative improvement in AUROC attributed to the features across all trees within the gradient boosting tree model. To focus more on the most impactful features (i.e., variable ranked among top 100) without losing information on the weaker features, we assigned a "soft" membership of a feature as how high up the rank is relative to top $s$ ($s = 100$) by applying an exponentially decreasing function to the original ranks ($r$), i.e., $f(r) = \exp\{-r/s\}$. We used SHAP values to evaluate the marginal effects of the shared top important variables of interests[33]. Specifically, the SHAP values evaluated how the logarithmic odds ratio changed by including a factor of certain values for each individual patient. The SHAP values not only captured the global patterns of effects of each factor but also demonstrated the patient-level variations of the effects. We also estimated 95% bootstrapped confidence intervals of SHAP values for each selected feature based on 100 bootstrapped samples.

**Adjusted maximum mean discrepancy (adjMMD)**. MMD has been widely used in transfer learning studies for maximizing the similarity among distributions of different domains[34,35]. Here we used MMD to measure the similarities of distributions for the same feature between training and validation sites. More specifically, let $\{X\}_1^K$ be $K$ random variables selected by the training model with their corresponding weights $\{w\}_i^K$ proportional to their "gains", where $X_k = [x_1, \dots, x_{n_k}]$ which are independent identical distributed (i.i.d.) samples collected from an underlying distribution $\mathbb{P}$. After we collect data from a validation site, we can observe the corresponding $k$ variables .., where $\tilde{X}_k = [\tilde{x}_1, \dots, \tilde{x}_{m_k}]$ which are independent identical distributed (i.i.d.) samples collected from an underlying distribution $\tilde{\mathbb{P}}$. It is intuitive to assume that $X$ and $\tilde{X}$ are defined on the same topological space $\mathcal{X}^K$. Then for each $k$, an unbiased estimator of individual MMD is defined as:

$$\widehat{\mathrm{MMD}}_k = \frac{1}{n_k(n_k - 1)} \sum_{j=1}^{n_k} \sum_{i \neq j}^{n_k} \kappa(x_i, x_j) + \frac{1}{m_k(m_k - 1)} \sum_{j=1}^{m_k} \sum_{i \neq j}^{m_k} \kappa(\tilde{x}_i, \tilde{x}_j) - 2 \frac{1}{n_k m_k} \sum_{j=1}^{m_k} \sum_{i=1}^{n_k} \kappa(x_i, \tilde{x}_j) \tag{1}$$

where $\kappa(x, y)$ is the kernel function with Gaussian kernel being the common choice.

However, the classic MMD calculation does not account for the effect from missing variables and is not even calculatable when certain continuous variable is completely missing. Thus, we made an adjustment to the metric to (1) enable the MMD calculation when a continuous variable is missing completely; (2) penalize the "missingness" harder than distribution discrepancies. To achieve that, we intentionally imputed the "missingness" with values that could make the $\tilde{\mathbb{P}}$ estimation significantly

deviate from $\mathbb{P}$, i.e., a single or range of values that is very unlikely to be observed under $\mathbb{P}$. For example, we chose to randomly sample values that is less than $\bar{X}_k - z_{0.99}s/\sqrt{n_k}$ or greater than $\bar{X}_k + z_{0.99}s/\sqrt{n_k}$, where $z_{0.99}$ denotes the 99th percentile value under a standard normal distribution. Or more generally, we used kernel density estimation and sample values from the bottom and top 1 percentile of the approximated underlying distribution which could not necessarily be normal.

To demonstrate the robustness of adjMMD, we repeated the experimental steps (described in Experimental Design) (a) through (e) but using LASSO and limited-GBM models on one prediction task (48-h prediction of moderate-to-severe AKI), and further generalized the linear regression model by fitting a mixed-effect model with random slopes at different derivation site level among data generated from multiple derivation sites and models. More specifically,

$$y_{ijk} = \beta_{0ij} + \beta_{1ij}x_{ijk} + \epsilon_{ijk};$$
$$\beta_{0ij} = \beta_0 + \epsilon_{0j}, \beta_{1ij} = \beta_1 + \epsilon_{1i} + \epsilon_{1j} \qquad (2)$$
$$\epsilon_{ijk} \sim N(0, \sigma I), \epsilon_{0ij} \sim N(0, \sigma_i I), \epsilon_{1i} \sim N(0, \sigma_{1i} I), \epsilon_{1j} \sim N(0, \sigma_{1j} I)$$

where the fixed effects and $\hat{\beta}_1$ estimate the expected change of $\Delta$AUC with a unit change in adjMMD, and $\hat{\sigma}_{1i}, \hat{\sigma}_{1j}$ estimates the expected variations of $\beta_1$ due to varying derivation site and models. (2) can be fitted by maximizing restricted maximum likelihood with all pairs of adjMMD and $\Delta$AUC generated from the experiments. The sample-agnostic test was performed by deriving similar linear equation as in (2) but only using data generated from one derivation site (could be across different models to account for potential variations) and validating it on the remaining sites. The model-agnostic test, on the other hand, was performed by deriving a similar linear equation as in (2) but only using data generated from one model (could be across different sites to account for potential variations) and validating it on the other models.

**Reporting summary**. Further information on research design is available in the Nature Research Reporting Summary linked to this article.

## Data availability

The clinical data used for training and validation in this study is not publicly available and restrictions apply to its use. The de-identified multi-center CDM datasets are managed in an Oracle database on a secure server, and may be available from the Greater Plains Collaborative clinical data network, subjective to individual institution's and network-wide data governance and ethical approvals.

## Code availability

We developed generic SQL codes against PCORnet CDM to extract data from Oracle database (12c) where multi-center dataset is stored, and open-source R libraries (e.g. xgboost) to conduct our experiments. Implementation details and code are available at https://github.com/kumc-bmi/AKI_CDM to allow independent replication.

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

## Acknowledgements

This research was supported in part by NIH/NIDDK under award number R01DK116986, PCORnet Kidney Health CRG Pilot Award, PCORI grant CDRN-1306-04631 and NIH/NCATS Clinical Translational Science Award (CTSA) grant UL1TR002366. YH was supported by the Major Research Plan of the National Natural Science Foundation of China (Key Program, Grant No. 91746204), the Science and Technology Development in Guangdong Province (Major Projects of Advanced and Key Techniques Innovation, Grant No.2017B030308008), and Guangdong Engineering

Technology Research Center for Big Data Precision Healthcare (Grant No.603141789047). SS was partially supported by the Blue KC Outcome Research Grants (No. 0925-0001). The content is solely the responsibility of the authors and does not represent the official views of the funders. We are also grateful for the support of participating GPC site principal investigators: Bradley Taylor from Medical College of Wisconsin, Tim Imler from Marshfield Clinic Research Institute, Abu Mosa from the University of Missouri, James McClay from the University of Nebraska Medical Center, Lindsay Cowell from the University of Texas Southwestern, and all sites' project managers and analysts for contributing their CDM data.

## Author contributions

M.L. and Y.H. initiated the project and the collaboration. M.L., Y.H., A.Y., and J.K. designed the overall study. L.W. coordinated data sharing as the overall G.P.C. principal investigator. X.S. and M.L. developed the training and testing setup. X.S. extracted the study cohort, cleaned up the data, and performed all experiments. A.Y., J.K., M.M., and S.S. contributed their clinical expertise in analyzing results. X.S. and M.L. wrote the paper, with revision advices provided by Y.H., A.Y., J.K., L.W., M.M., and S.S.

## Competing interests

The authors declare no competing interests.
