## [Peer Review File · Nature Communications]

Reviewers' Comments:

Reviewer #1:

Remarks to the Author:

MAJOR COMMENTS

1. What I like about this paper include the "white box" transparency. My personal bias (as a practicing physician who regularly takes care of patients with acute kidney injury) is that artificial intelligence models will be more readily accepted by physicians and implemented into clinical care if they "make sense."

2. My concern with this entire literature has to do with the fact that I am not sure what clinicians are supposed to do with this type of information. There are no therapies which have been proven in randomized controlled trial to reduce risk of development of acute kidney injury in high risk patients (even if we were to be given a 24 to 48-hour advanced warning).

I know that notwithstanding this, some papers have gotten into very high impact factor journals and received a lot of press--such as reference 9: Tomašev, N., et al. A clinically applicable approach to continuous prediction of future acute kidney injury. *Nature* 572, 116-119 (2019)(which is a more black box approach). So I do not know if Song et al should be singled out and "penalized" for this compared with other research groups. But this larger consideration does dampen my enthusiasm.

3. On the other hand, there have not been that many major improvements in therapy for AKI in recent decades and this current approach leveraging state-of-the-art AI methods is clearly innovative. One can argue that having these better risk prediction models will help us launch clinical trials which can be conducted efficiently to come up with evidence-based practices to reduce development of frank acute kidney injury.

In order to break out of the Catch-22 (no trials so identifying high risk patients is futile < can't do trials easily without identifying high risk patients), one needs to start somewhere.

4. Another thing I like about this paper is that the authors explicitly considered leaving Cr and BUN out of the model and showed that it still performed pretty well. As far as I can tell, this was not done in reference 9. Since the definition of acute kidney injury is a rise in serum Cr of a certain degree, prior levels of Cr and Cr trajectory (and levels/trajectories of things which closely track Cr, such as BUN and Phos) of course contribute to a very high AUC.

Rising Cr is not just a predictor (like sepsis), it is "in the causal pathway" so to speak.

5. I also like the fact that the authors separately looked at stages 2 and 3 AKI. Many cases of stage I AKI with 0.3 mg/dl rise in Cr over 48 hrs have minimal clinical significance (see also PubMed 26336912 click).

6. I think this paper makes clear contributions in terms of pointing out important transportability issues to consider for AKI prediction models.

Leveraging the GPC PCORnet Clinical Data Research Network is a strength—in terms of what is currently feasible, "real life" applications and being more generalizable than the VA data (ref 9) or single center studies (e.g. ref 8). The VA EMR is very homogenous even though it spans many hospitals/clinics.

7. Identifying sources of performance heterogeneity is valuable.

8. adjMMD is innovative.

MINOR COMMENT

9. One can quibble with some of the clinical decisions that were made e.g. "Burn patients were also excluded since serum creatinine becomes a less reliable tool in assessing renal function during hypermetabolic phase." There are numerous other situations in which Cr performance is also problematic, such as cirrhosis, amputation, or muscle wasting diseases which lead to changes in Cr production. But this is not a major point as burn patients are not large in number.

Reviewer #2:

Remarks to the Author:

This paper describes a study with three major contributions: (1) a tree-based GBM to predict the development of AKI across all hospitalized patients with a lead time of 48 hours; (2) evaluate the transportability of the model in other hospitals with and without re-fitting; (3) a novel method to predict transportability of machine learning models named adjMMD, that can be used to infer the model performance deteriorations.

The paper is well written and easy to read. I think the authors did a lot of work to prove their ideas and results seem promising and generalizable for the community.

I have a few major comments that I think should be added to strengthen the manuscript.

(1) The GTM is not compared with any algorithm and I recommend to report results also for logistic regression as baseline. This would help to support some considerations made in text, that are very general but only based on one method. In particular, I would be curious to see if the deterioration with LR is proportional or maybe the choice of the model plays a factor in the external performances.

(2) The utility of adjMMD is not clear in my opinion. Experiments show that refitting the model in the external sites work better than just using the pre-trained models. So, what is the point of knowing the deterioration in advance? If the model is re-trained in the new site, maybe more features specific to new site can be added to the model, and performance might be similar. I would be curious to have a clarification about this. In my opinion, it could be more useful if adjMMD can be used to decide if directly applying the pre-trained model to new data or if adjusting the pre-trained model by tuning it with the data of the new site (as you would do, for example, with neural networks). I would like the author to address better this issue and clarify more the usefulness of adjMMD in practice, highlighting some best practices depending on the score (Should we use the model as it is? Should we consider another model for the new site? Should we re-train? And so on).

(3) The description of how the coefficients for the linear equation to compute performance changes of a model (e.g., -0.018 and 0.344) is not very clear in my opinion and need to be improved in the text to facilitate the reader.

Reviewer #3:

Remarks to the Author:

Thank you for the opportunity to review this interesting manuscript by Song et al. which investigates the transportability of an acute kidney injury prediction model. In this work, the investigators test the transportability of a machine learning model across six different health systems. Their results suggest that model performance degrades during external validation due to heterogeneity of risk factors between sites. They further develop a method to predict transportability machine learning models. External generalizability of machine learning models is

an important area of research, and efforts to understand and improve transportability are critical. I have several comments aimed at improving this well-written manuscript, including additional experiments to validate the key results

Major comments

1. The most novel aspect of this work is the adjMMD and its strong association with decrement in AUC. However, the experiments performed thus far are not convincing regarding its association with the change in AUC. More experiments should be performed, including: 1. Deriving models with each other the other six sites being the derivation site to confirm that this relationship still holds; 2. Testing this association with another commonly used machine learning model (e.g., elastic net); 3. Testing this association with a trimmed list of variables (e.g., age, BMI, vitals, CBC, BMP, and trends) that are likely to be collected at all hospitals.
2. More details are needed in the methods regarding some of the experiments, in particular the refitting during the transportability experiments.

Minor comments

Abstract

1. The authors have not proven in this work that cross-site performance deterioration is unavoidable in all cases, just in the clinical scenario they studied, so this wording should be softened (e.g., cross-site deterioration in performance is likely, or something similar).

Introduction

1. This is well-written overall. One minor grammatical edit is that there are missing commas after "i.e." and "e.g."
2. Also, it should be "discrete-time" survival analysis (not "discrete survival analysis").
3. It would be nice to add more information regarding why transportability is important. What are the costs/harms of low transportability? Since the paper is about studying and improving transportability, a stronger argument regarding its importance is needed. To shorten the introduction after this addition, the details about PCORnet can be shortened, as there is some overlap with what is included in the Methods.

Methods

1. Were ICD and CPT codes included in the models? If so, were these codes from prior to the admission or from the current admission? Codes from the current admission are often not available in real-time due to delays in billing filing, so these variables should not be included in the model.
2. How were different medications from the same class, which could be due to different formularies at different sites, handled?
3. Can you clarify whether patients were censored at 7 days if still in the hospital? If so, did you still test the model in patients with LOS of >7 days or was the model not validated in those patients? This should be added to the limitations, as LOS>7 days is not uncommon, especially in critically ill patients.
4. As above, discrete-time survival analysis is the standard terminology (e.g., Singer & Willett, 1993).
5. The details about how the data were split into different cohorts for training and validation should be included in the Methods.
6. Additional information regarding how the refitting procedure during the transportability tests are needed. For example, what proportion of the external data was used to refit the model vs. test the model? Did the refitting combining the training cohort with part of the external site's data?
7. The adjMMD metric's association with AUC drop as well as the linear equation proposed in the paper needs to be validated. One easy way to validate it would be to perform five more rounds of transportability tests, with each of the other individual sites acting as the derivation site.
8. How would a less complex model such as elastic net, which can perform variable selection and may be less likely to overfit than gradient-boosted trees, perform in the transportability tests? Would the same relationship occur between adjMMD and drop in AUC?

9. Similarly, would adjMMD have the same association with drop in AUC if a simpler model using GBM were developed using only commonly collected data (e.g., age, BMI, vital signs, BMP, CBC, and trends)?

Results

1. Well-written, with suggested clarifications as noted in Methods.

Discussion

1. Although the CDM allowed for the inclusion of additional variables, is this really an interpretable model when applied to an individual patient? Could clinicians really understand a model with 30,000 predictor variables, with likely thousands of them combining to form small, but meaningful effects on predictions?

2. If the new hospital data are available, why not just refit the data using those data as a matter of practice as opposed to calculating transportability metrics first? Additional discussion regarding this and the advantages of using adjMMD should be discussed.

We sincerely appreciate the three reviewers for their constructive comments on our manuscript. We have made every attempt to fully address these comments in the revised manuscript. Below, we outline how we handled each reviewer's comments.

REVIEWER COMMENTS

Reviewer #1 (Remarks to the Author):

MAJOR COMMENTS

1. What I like about this paper include the “white box” transparency. My personal bias (as a practicing physician who regularly takes care of patients with acute kidney injury) is that artificial intelligence models will be more readily accepted by physicians and implemented into clinical care if they “make sense.”

RESPONSE: We completely agree with the reviewer. Some of the authors are also physicians who routinely care for these patients and encouraged this approach for exactly this reason.

2. My concern with this entire literature has to do with the fact that I am not sure what clinicians are supposed to do with this type of information. There are no therapies which have been proven in randomized controlled trial to reduce risk of development of acute kidney injury in high risk patients (even if we were to be given a 24 to 48-hour advanced warning). I know that notwithstanding this, some papers have gotten into very high impact factor journals and received a lot of press—such as reference 9: Tomašev, N., et al. A clinically applicable approach to continuous prediction of future acute kidney injury. *Nature* 572, 116-119 (2019) (which is a more black box approach). So I do not know if Song et al should be singled out and “penalized” for this compared with other research groups. But this larger consideration does dampen my enthusiasm.

RESPONSE: The reviewer is correct that new interventions for AKI have been slow to be developed. However, much of AKI is iatrogenic (medications, procedures) and these are under physician control both in terms of timing and selection (and for medications, dose). Much of the KDIGO clinical practice guideline concerns these aspects and moving attention to these issues up is likely to be beneficial. This combined with general (though unproven) measures such as optimizing fluid and hemodynamic status is currently the best care we have. We recognize that some will say that we should do all these things for all patients, but the reality is that this is neither possible nor optimal. In some patients the risk of AKI is low and drugs like NSAIDS are very appropriate (especially to avoid narcotics) while in other patients, the risks may outweigh the benefits. The timing of procedures (e.g. surgery) can be impacted and the use of invasive hemodynamic monitoring can be selected based on patient risk.

There is some evidence that implementation of “care bundles” consisting of very general measures such as those described above, particularly if implemented early (e.g. in response to an electronic alert), is associated with improved outcomes, including recovery from AKI, length of stay, and mortality (PMID: 22067631, 27190331, 30089118).

We have added a paragraph to the Discussion on p. 17, before “Limitations and Future Work”.

3. On the other hand, there have not been that many major improvements in therapy for AKI in recent

decades and this current approach leveraging state-of-the-art AI methods is clearly innovative. One can argue that having these better risk prediction models will help us launch clinical trials which can be conducted efficiently to come up with evidence-based practices to reduce development of frank acute kidney injury. In order to break out of the Catch-22 (no trials so identifying high risk patients is futile < can't do trials easily without identifying high risk patients), one needs to start somewhere.

RESPONSE: We agree with the reviewer that this is another important application of these models.

4. Another thing I like about this paper is that the authors explicitly considered leaving Cr and BUN out of the model and showed that it still performed pretty well. As far as I can tell, this was not done in reference 9. Since the definition of acute kidney injury is a rise in serum Cr of a certain degree, prior levels of Cr and Cr trajectory (and levels/trajectories of things which closely track Cr, such as BUN and Phos) of course contribute to a very high AUC. Rising Cr is not just a predictor (like sepsis), it is “in the causal pathway” so to speak.

RESPONSE: Thank you. We completely agree.

5. I also like the fact that the authors separately looked at stages 2 and 3 AKI. Many cases of stage I AKI with 0.3 mg/dl rise in Cr over 48 hrs have minimal clinical significance (see also PubMed 26336912 click).

RESPONSE: Thank you. We completely agree. We have added this reference.

6. I think this paper makes clear contributions in terms of pointing out important transportability issues to consider for AKI prediction models. Leveraging the GPC PCORnet Clinical Data Research Network is a strength—in terms of what is currently feasible, “real life” applications and being more generalizable than the VA data (ref 9) or single center studies (e.g. ref 8). The VA EMR is very homogenous even though it spans many hospitals/clinics.

RESPONSE: Thank you. We completely agree.

7. Identifying sources of performance heterogeneity is valuable.

RESPONSE: Thank you.

8. adjMMD is innovative.

RESPONSE: Thank you.

MINOR COMMENT

9. One can quibble with some of the clinical decisions that were made e.g. “Burn patients were also excluded since serum creatinine becomes a less reliable tool in assessing renal function during hypermetabolic phase.” There are numerous other situations in which Cr performance is also problematic, such as cirrhosis, amputation, or muscle wasting diseases which lead to changes in Cr production. But this is not a major point as burn patients are not large in number.

RESPONSE: We agree.

Reviewer #2 (Remarks to the Author):

This paper describes a study with three major contributions: (1) a tree-based GBM to predict the development of AKI across all hospitalized patients with a lead time of 48 hours; (2) evaluate the transportability of the model in other hospitals with and without re-fitting; (3) a novel method to predict transportability of machine learning models named adjMMD, that can be used to infer the model performance deteriorations.

The paper is well written and easy to read. I think the authors did a lot of work to prove their ideas and results seem promising and generalizable for the community. I have a few major comments that I think should be added to strengthen the manuscript.

1. The GTM is not compared with any algorithm and I recommend to report results also for logistic regression as baseline. This would help to support some considerations made in text, that are very general but only based on one method. In particular, I would be curious to see if the deterioration with LR is proportional or maybe the choice of the model plays a factor in the external performances.

RESPONSE: Thank you for the suggestion. We have added experimental results for the 48-hour prediction of moderate-to-severe AKI using a LASSO logistic regression model in the Results section on page 5 and 9 as well as Supplemental Figures 4 and 8. The performance of LASSO is consistently worse than the GBT model in all internal, temporal and external validations. For the GBT model, the deterioration of AUROC is from 0.81 [95%CI 0.76-0.86](temporal validation) to 0.73 [95%CI 0.72 – 0.85] (median of external validation); while the LASSO model showed similar degree of deterioration from 0.78 [95%CI 0.73-0.84](temporal validation) to 0.68 [95%CI 0.67-0.69] (median of external validation).

2. The utility of adjMMD is not clear in my opinion. Experiments show that refitting the model in the external sites work better than just using the pre-trained models. So, what is the point of knowing the deterioration in advance? If the model is re-trained in the new site, maybe more features specific to new site can be added to the model, and performance might be similar. I would be curious to have a clarification about this. In my opinion, it could be more useful if adjMMD can be used to decide if directly applying the pre-trained model to new data or if adjusting the pre-trained model by tuning it with the data of the new site (as you would do, for example, with neural networks). I would like the author to address better this issue and clarify more the usefulness of adjMMD in practice, highlighting some best practices depending on the score (Should we use the model as it is? Should we consider another model for the new site? Should we re-train? And so on).

RESPONSE: We tried to clarify the usefulness of adjMMD by highlighting some practices depending on the score that the users can use in the Discussion section as follows. Since hospitals across the US and globally have different data/information infrastructure maturity and AI capability, adjMMD can be useful in different scenarios. When a prediction model is transported to a hospital with less mature clinical data warehouse (i.e. insufficient amount of quality data for building and validating an independent model), users can use the adjMMD score to decide if the externally trained model can be used in clinic as is with acceptable performance. Alternatively, if the target hospital is mature in both data and AI capability, adjMMD can then be used to decide if the model need to be updated with

varying amount of effort ranging from simple recalibration to model refitting on local data and even full model revision with the incorporation of new predictors. We also provided some implementation details in the Discussion section on page 16 on how adjMMD could be used in practice based the additional robustness experiments we performed.

3. The description of how the coefficients for the linear equation to compute performance changes of a model (e.g., -0.018 and 0.344) is not very clear in my opinion and need to be improved in the text to facilitate the reader.

RESPONSE: To clarify, on page 12, we have added more detailed description of the simple linear equation between adjMMD and Δ AUC. Key of the equation is the slope 0.344, which suggests that every 0.1 unit increase in the adjMMD would potentially lead to 0.0344 decrease in the target AUROC (or an expected AUROC of 0.77 when applying KUMC model to a new site).

Reviewer #3 (Remarks to the Author):

Thank you for the opportunity to review this interesting manuscript by Song et al. which investigates the transportability of an acute kidney injury prediction model. In this work, the investigators test the transportability of a machine learning model across six different health systems. Their results suggest that model performance degrades during external validation due to heterogeneity of risk factors between sites. They further develop a method to predict transportability machine learning models. External generalizability of machine learning models is an important area of research, and efforts to understand and improve transportability are critical. I have several comments aimed at improving this well-written manuscript, including additional experiments to validate the key results

MAJOR COMMENTS

1. The most novel aspect of this work is the adjMMD and its strong association with decrement in AUC. However, the experiments performed thus far are not convincing regarding its association with the change in AUC. More experiments should be performed, including: 1. Deriving models with each other the other six sites being the derivation site to confirm that this relationship still holds; 2. Testing this association with another commonly used machine learning model (e.g., elastic net); 3. Testing this association with a trimmed list of variables (e.g., age, BMI, vitals, CBC, BMP, and trends) that are likely to be collected at all hospitals.

RESPONSE: Thank you for the suggestions. We have run multiple additional experiments as below and inserted the results into the revised manuscript (page 13 of Results, page 16 of Discussion and page 22 of Method).

Additional Experiment #1: Using each of the six sites as a derivation site to derive a GBT model and calculated adjMMD for 48-hour prediction of moderate-to-severe AKI, to confirm that the positive relationship between adjMMD and Δ AUC still holds.

Additional Experiment #2: Testing the association between adjMMD and Δ AUC with LASSO, i.e. training a LASSO model using each of the six sites as a derivation site (same as in Additional Experiment #1 except with LASSO).

Additional Experiment #3: Built and validated a GBT model with a trimmed list of variables from the KUMC source dataset that are collected and presented at all participating sites (i.e. the “limited-GBT”), and then tested the association between adjMMD and Δ AUC using KUMC as the derivation site and remaining sites as testing (same as in Additional Experiment #1 except with the limited-GBT model).

Using data generated from these three additional experiments, we first fit a mixed-effect model: Δ AUC, $y = -0.014 + 0.257 x$ with a 95% CI for the slope to be [0.08, 0.44], which confirm the significantly strong positive correlation between adjMMD and Δ AUC. We further performed two agnostic tests: a) *sample-agnostic tests*, where we derived the linear equation from the same derivation site (*same site/sample*) and evaluated how well it fits to the other sites (*different site/sample*); and b) *model-agnostic tests*, where we derived the linear equation from the same model (*same model*) and evaluated how well it fits to the other models (*different model*). These two models showed no significant differences in residuals sum of square (RSS) between each comparison groups, which supports that adjMMD can be used as a robust indicator of performance change over varying training set, model and feature space.

2. More details are needed in the methods regarding some of the experiments, in particular the refitting during the transportability experiments.

RESPONSE: We have added more details about experimental details in the “Experimental Design” paragraph within Method section on page 19.

MINOR COMMENTS

Abstract

1. The authors have not proven in this work that cross-site performance deterioration is unavoidable in all cases, just in the clinical scenario they studied, so this wording should be softened (e.g., cross-site deterioration in performance is likely, or something similar).

RESPONSE: Thank you for the suggestion. We have made the change in the Abstract.

Introduction

1. This is well-written overall. One minor grammatical edit is that there are missing commas after “i.e.” and “e.g.”

RESPONSE: We have inserted the missing commas throughout the manuscript.

2. Also, it should be “discrete-time” survival analysis (not “discrete survival analysis”).

RESPONSE: We have made the corresponding changes throughout the manuscript.

3. It would be nice to add more information regarding why transportability is important. What are the costs/harms of low transportability? Since the paper is about studying and improving transportability, a stronger argument regarding its importance is needed. To shorten the introduction after this addition,

the details about PCORnet can be shortened, as there is some overlap with what is included in the Methods.

RESPONSE: We have inserted a paragraph that highlights the clinical importance of model transportability in the Introduction on page 2.

Methods

1. Were ICD and CPT codes included in the models? If so, were these codes from prior to the admission or from the current admission? Codes from the current admission are often not available in real-time due to delays in billing filing, so these variables should not be included in the model.

RESPONSE: Both ICD and CPT codes were included in the model. The ICD codes included were for medical diagnoses from past encounters. The CPT codes were used to capture procedures performed during the current hospital stay. When the prediction model is implemented in the hospital, we envisioned that the procedure information would be captured as real-time orders or entries in the EMR. We had to use delayed CPT billing codes as a proxy for capturing procedures in this study because the current PCORnet CDM does not specify collection of procedure orders which is readily available in real-time in the hospital EMR. We have inserted this into the limitations.

2. How were different medications from the same class, which could be due to different formularies at different sites, handled?

RESPONSE: RxNorm was used to normalize the medications at the ingredient level.

3. Can you clarify whether patients were censored at 7 days if still in the hospital? If so, did you still test the model in patients with LOS of >7 days or was the model not validated in those patients? This should be added to the limitations, as LOS>7 days is not uncommon, especially in critically ill patients.

RESPONSE: Yes, patients were censored at 7 days if they are still in the hospital. We did not test the model in patients with length of stay greater than 7 days. We agree with the reviewer that many critically ill patients' hospital length of stay is greater than 7 days, so we have added this into the Limitation and Future Work section.

4. As above, discrete-time survival analysis is the standard terminology (e.g., Singer & Willett, 1993).

RESPONSE: We have inserted the citation into the Methods section.

5. The details about how the data were split into different cohorts for training and validation should be included in the Methods.

RESPONSE: We have added more relevant details about experimental details in the "Experimental Design" paragraph within Methods section on page 19.

6. Additional information regarding how the refitting procedure during the transportability tests are

needed. For example, what proportion of the external data was used to refit the model vs. test the model? Did the refitting combining the training cohort with part of the external site's data?

RESPONSE: We have added more relevant details about experimental details in the "Experimental Design" paragraph within Methods section on page 19.

7. The adjMMD metric's association with AUC drop as well as the linear equation proposed in the paper needs to be validated. One easy way to validate it would be to perform five more rounds of transportability tests, with each of the other individual sites acting as the derivation site.

RESPONSE: We have run multiple additional experiments and inserted the results into the revised manuscript (page 13 of Results, page 16 of Discussion and page 22 of Method). One of the experiments is to use each of the six sites as a derivation site to derive a different GBT model and calculated adjMMD for 48-hour prediction of moderate-to-severe AKI, to confirm that the positive relationship between adjMMD and Δ AUC still holds. Results from the additional experiments supported the robustness of adjMMD.

8. How would a less complex model such as elastic net, which can perform variable selection and may be less likely to overfit than gradient-boosted trees, perform in the transportability tests? Would the same relationship occur between adjMMD and drop in AUC?

RESPONSE: We have run multiple additional experiments and inserted the results into the revised manuscript (page 13 of Results, page 16 of Discussion and page 22 of Method). One of the experiments is to test the association between adjMMD and Δ AUC with LASSO model, i.e. training a LASSO model using each of the six sites as a derivation site (same as in Additional Experiment #1 except with LASSO). Results from the additional experiments supported the robustness of adjMMD.

9. Similarly, would adjMMD have the same association with drop in AUC if a simpler model using GBM were developed using only commonly collected data (e.g., age, BMI, vital signs, BMP, CBC, and trends)?

RESPONSE: We have run multiple additional experiments and inserted the results into the revised manuscript (page 13 of Results, page 16 of Discussion and page 22 of Method). One of the experiments is to develop another GBT model with a trimmed list of variables from the KUMC source dataset that are collected and presented at all participating sites (i.e. the "limited-GBT"), and then test the association between adjMMD and Δ AUC. The additional experiment has supported the robustness of adjMMD.

Results

1. Well-written, with suggested clarifications as noted in Methods.

RESPONSE: Thank you. We have made suggested clarifications in Methods as detailed above.

Discussion

1. Although the CDM allowed for the inclusion of additional variables, is this really an interpretable

model when applied to an individual patient? Could clinicians really understand a model with 30,000 predictor variables, with likely thousands of them combining to form small, but meaningful effects on predictions?

RESPONSE: For each patient, the model can make individualized risk prediction and because of the white-box algorithm utilized we are able to show the most important factors that affected the prediction for an individual patient. Figure 2 illustrated marginal plots of the top 10 important variables for predicting moderate to severe AKI in 48 hours aggregated over the population. To show effects of the most important predictors on individual AKI risk in the revised manuscript, we randomly selected some patients and illustrated their estimated logarithmic odds ratios in an interactive dashboard at https://sxinger.shinyapps.io/AKI_ishap_dashbd/. A paragraph describing this dashboard was inserted in the Results section on page 7.

2. If the new hospital data are available, why not just refit the data using those data as a matter of practice as opposed to calculating transportability metrics first? Additional discussion regarding this and the advantages of using adjMMD should be discussed.

RESPONSE: Since hospitals across the US and globally have different data/information infrastructure maturity and AI capability, refitting model or training a new model with local data is not always feasible. Some hospitals may not have enough good quality data for that purpose. We envisioned that adjMMD can be useful in different scenarios. When a prediction model is transported to a hospital with less mature clinical data warehouse (i.e. insufficient amount of quality data for building and validating an independent model), users can use the adjMMD score to decide if the externally trained model can be used in clinic as is with acceptable performance. Alternatively, if the target hospital is mature in both data and AI capability, adjMMD can then be used to decide if the model need to be updated with varying amount of effort ranging from simple recalibration to model refitting on local data and even full model revision with the incorporation of new predictors. We have inserted a paragraph in the Discussion section to highlight the usefulness of adjMMD.

Reviewers' Comments:

Reviewer #2:

Remarks to the Author:

The authors clarified all my concerns and the paper reads much better now. I don't have any other comment and I think it is ready for publication.

Reviewer #3:

Remarks to the Author:

Thank you for the opportunity to review this revised submission by Song et al. The authors have done an excellent job responding to all my comments, and their additional experiments have confirmed the value of their novel adjMMD metric.

My only comment is that the authors should note the recent validation of the GBM AKI model developed by Koyner et al.: <https://www.ncbi.nlm.nih.gov/pmc/articles/PMC7420241/>, which is relevant to the current manuscript.

REVIEWER COMMENTS

Reviewer #2 (Remarks to the Author):

The authors clarified all my concerns and the paper reads much better now. I don't have any other comment and I think it is ready for publication.

RESPONSE: Thank you very much.

Reviewer #3 (Remarks to the Author):

Thank you for the opportunity to review this revised submission by Song et al. The authors have done an excellent job responding to all my comments, and their additional experiments have confirmed the value of their novel adjMMD metric.

My only comment is that the authors should note the recent validation of the GBM AKI model developed by Koyner et al.: <https://www.ncbi.nlm.nih.gov/pmc/articles/PMC7420241/>, which is relevant to the current manuscript.

RESPONSE: We truly appreciate your review comments that helped to improve the paper. We have added a sentence in the Introduction citing the newly published paper by Koyner et al.